# Clinical Significance of Diabetes-Mellitus-Associated Antibodies in Rheumatoid Arthritis

**DOI:** 10.3390/cells11223676

**Published:** 2022-11-18

**Authors:** Malin C. Erlandsson, Mahomud Tuameh, Elin Jukic Huduti, Sofia T. Silfverswärd, Rille Pullerits, Maria I. Bokarewa

**Affiliations:** 1Department of Rheumatology and Inflammation Research, Institute of Medicine, The Sahlgrenska Academy at University of Gothenburg, Guldhedsgatan 10A, SE-41345 Gothenburg, Sweden; 2Rheumatology Clinic, The Sahlgrenska University Hospital, Region of West Götaland, SE-41345 Gothenburg, Sweden; 3Department of Clinical Immunology and Transfusion Medicine, The Sahlgrenska University Hospital, SE-43145 Gothenburg, Sweden

**Keywords:** rheumatoid arthritis, diabetes mellitus, autoantibodies, prospective follow-up

## Abstract

Rheumatoid arthritis (RA) is a canonical autoimmune disease that shares numerous risk factors with diabetes mellitus (DM). The production of autoantibodies is a characteristic feature in both diseases. To determine the frequency and specificity of DM-related antibodies (DMab) in RA patients and to study whether DMab associates with new DM cases in RA patients, we measured DMab defined as IgG against glutamic acid decarboxylase (GADA), tyrosine phosphatase (IA2-ab), and zinc transporter (ZnT8-ab) in a cohort of 290 RA patients (215 women and 75 men, median disease duration 11 years). Of those, 21 had a DM diagnosis at baseline. The development of new DM cases and mortality were traced in a 10-year prospective follow-up. Predictive analyses for DM and mortality were carried out by the Mantel–Cox regression. We found that 27 of the patients (9.3%) had DMab, equally often men and women. The presence of DMab was more frequent in patients with DM (*p* = 0.027. OR 4.01, 95%CI [1.20; 11.97]), suggesting their specificity for the disease. Men had more prevalent incidental DM at the baseline (12% vs. 5%, *p* = 0.030) and among the new DM cases (*p* = 0.012. HR 6.08, 95%CI [1.57; 25]). New DM developed equally frequently in DMab-positive and DMab-negative patients. DM, but not DMab, significantly increased the estimated mortality rate in RA patients (*p* = 0.021, OR 4.38 [1.2; 13.52]). Taken together, we conclude that DMab are associated with DM in RA patients, but they are not solely enough to predict disease development or mortality in those patients.

## 1. Introduction

Rheumatoid arthritis (RA) and diabetes mellitus (DM) are complex and heterogenous autoimmune diseases. These conditions share numerous risk factors that contribute to their development, including metabolic alterations [1,2], inflammation [3,4], and autoimmune reactions against self-antigens, which result in autoantibody production [5,6,7]. Both conditions are widely recognized as a major contributor to morbidity and mortality in the western world, not least due to their strong associations with cardiovascular disease [8,9,10].

Over the last decades, new effective treatments have become increasingly available, improving the prognosis and quality-of-life of patients with RA and DM. However, the affected individuals with both diseases still have a reduced life expectancy compared with the general population [9,11]. Several studies investigated the prevalence of DM in RA and demonstrated an increased rate of incidental DM in RA compared with controls [12,13,14].

In RA patients, the prevalence of DM was investigated in multiple case-controlled studies, and variable outcomes were reported (meta-analysis, [15,16]). The Nurses’ Health Study in the US observed no difference in risk of DM among the women with RA and without RA [17,18]. The cohort study from Canada reported the higher risk of type-two DM (T2D) in RA patients compared with controls [19], which stayed after adjustments for age, sex and glucocorticoid treatment. In contrast, the Outcome of Rheumatoid Arthritis Longitudinal Evaluation and San Antonio Heart Study demonstrated a higher frequency of T2D among RA patients; however, the difference was eliminated after adjustment for age [20]. Analogously, a British cohort study reported that the increased incidence of DM in RA patients leveled-off after adjustments were made for obesity, alcohol consumption, and smoking [12]. Two independent national-wide studies from Taiwan presented results in favor of higher T2D risk in RA patients [21,22]. In one of these studies, the risk was significantly increased in females [21], but in the other one in the risk was higher for men [22]. In total, the studies emphasized a connection between the development of DM in RA patients and in such predisposing risk factors as obesity and glucocorticoid medication characteristic for T2D. The predictive role of diabetes mellitus antibodies (DMab) for the development of the disease in RA adults attracted only a limited attention.

The self-antigen specific immune response classifies both RA and DM as autoimmune diseases [13] which are connected by common molecular pathways [23]. It is estimated that 50–70% of RA patients produce autoantibodies specific for this disease, which contribute to joint inflammation by promoting complement activation, inducing osteoclast activation, and increasing the process of neutrophil extracellular traps formation known as NETosis, which in turn trigger further inflammatory mechanisms [24]. RA-specific antibodies (RAab) are valuable for both the early diagnosis and for the long-term prognosis with respect to treatment response, skeletal damage progress, and rate of extra-articular complications [24,25].

By antigen targets, RAabs are described as a rheumatoid factor (RF) that bind to the Fc fragment of aggregated IgG [25] and antibodies against citrullinated peptides (ACPA) [26]. Among the most important antigen targets in DM are the 65 kD isoform of the enzyme glutamic acid decarboxylase (GAD65), an enzyme synthesized mainly in the islet cells of pancreas; tyrosine-phosphatase-related islet cell-specific antigen two (IA2), zinc transporter eight (ZnT8), and the pancreatic islet cell autoantigen (ICA) [2,8].

In the clinical setting, DM is commonly separated into two major forms, type-one diabetes (T1D) defined by the immune-mediated destruction of insulin-producing beta-cells in the pancreas, causing insulin deficiency, and T2D, traditionally explained by metabolic determinants and the low sensitivity of cells and tissues to insulin, the phenomenon called peripheral insulin resistance [8]. T1D affects approximately 10–15% of all patients, consequently making T2D by far the most common variant [27].

It is estimated that about 90% of patients with T1D have autoantibodies, which are often present years before diagnosis and act as predictive biomarkers for the disease. The number of antibodies, along with the type, titers, affinity, and age of seroconversion are all associated with the development of clinically distinguishable DM [27,28]. Recently, a variant called latent autoimmune diabetes in adults (LADA) was defined. LADA patients must have at least one diabetes-mellitus autoantibody (DMab), debut at an age over 30, and have no need for insulin treatment during the first 6 months after diagnosis [29]. Typically, LADA patients develop a fast decline in beta-cell function. In a prospective study, a UK research group investigated the need for insulin treatment among T2D patients and found that 84–94% of DMab-positive patients needed insulin 6 years after diagnosis, compared with only 14% of DMab-negative patients [30]. The investigation of DMab in T2D patients resulted in a common prediction of approximately 10% [29], which doubled the prevalence of DM with an autoimmune cause. Additionally, the autoimmune component opened for a new treatment of LADA patients including drugs for targeting systemic inflammation and immune cells to improve the metabolic profile, which appeared as a good addition to traditional glucose-lowering drugs [8,31].

In this study, we asked whether the autoimmune background of RA increases the risk of LADA, and whether the presence of DMab predicts the development of DM in a cohort of patients with RA.

## 2. Materials and Methods

### 2.1. Study Design and Patient Population

This longitudinal observational study included 290 patients with established RA (215 female and 75 male) enrolled at the Rheumatology Clinic of Sahlgrenska University Hospital, Gothenburg, and at the Northern Älvsborg County Hospital, Uddevalla, between 2011 and 2018 (Figure 1A).

At inclusion, a standardized physical examination by a rheumatologist was performed on all RA patients. The clinical and laboratory parameters of the disease were collected. To be included, the participants needed to fulfill the European League Against Rheumatism and American College of Rheumatology’s 2010 revised criteria for RA [32]. The exclusion criteria were other serious physical or psychiatric illnesses that would hinder participation in either the physical examination, blood sampling, or the patient interview, which was performed in Swedish. At inclusion, all patients underwent a clinical assessment of disease activity in 28 joints and completed the Stanford Health Assessment Questionnaire disability index (HAQ) [33]. The disease-activity scores were calculated based on the examinations of 28 tender and swollen joints (DAS28) and the erythrocyte sedimentation rates (ESR). A DAS28 below 2.6 indicated remission [34]. The body mass indexes (BMI) were calculated based on height and weight and are expressed in kg/m^2^.

### 2.2. Prospective Follow-Up

The follow-up was conducted during the years 2021 and 2022. The median follow-up time was 111 months (range 33–122 months), a total of 2400 patient/years (Figure 1B). Men were contacted for structured phone-interviews, which included questions regarding DM diagnosis and treatment in September 2021. For the female participants, structured interviews were performed during 2018, and were further completed through the evaluation of hospital medical records in February 2022. The variables extracted from the patient interviews included the diagnosis of DM, cardiovascular events, and new medications.

The medical records were used to confirm new cases of DM. Additionally, the digital prescription list from the medical records was examined to identify anti-diabetic drugs. The participant was documented to have DM if a DM diagnosis or anti-diabetic drug prescription was found in the medical records. These new cases of DM were further confirmed in the Swedish National Patient Register.

### 2.3. Ethics

The study was approved by the Swedish Ethical Review Authority (Dnr 659-11) and was conducted in accordance with national regulations and the International Conference on Harmonization Good Clinical Practice requirements, based on the Declaration of Helsinki. All patients provided written informed consent before being subjected to any study-related procedures. The study is registered at ClinicalTrials.gov with ID NCT03449589.

### 2.4. Blood Sampling and Measurements

Blood samples were collected between 7 and 10 o’clock in the morning after overnight fasting. The blood was obtained from the cubital vein into vacuum containers, and serum and plasma samples were stored at −70 °C until use.

### 2.5. Measurement of Autoantibodies

DM autoantibodies (DMab) were measured in the patient serums according to the manufacturer’s instructions. The anti-glutamic acid decarboxylase IgG antibodies (GADA) and the anti-tyrosine phosphatase IgG antibodies (IA2-ab) were measured using ELISA kits from Euroimmun^®^ (Lübeck, Germany). The anti-zinc transporter 8 IgG antibodies (ZnT8-ab) were analyzed using an ELISA kit from RSR limited (Cardiff, UK). The results are expressed in international units (IU/mL). The cut-off limits recommended by the manufacturers and validated at the Laboratory of Clinical Immunology at Sahlgrenska University Hospital were used to identify positive samples. The samples with the levels of antibodies above 5 IU/mL for GADA, 10 IU/mL for IA2-ab, and 15 IU/mL for ZnT8-ab were designated as positive [35].

Rheumatoid factor (RF) and anti-citrullinated protein antibodies (ACPA) were measured in the serum samples at the accredited Laboratory of Clinical Immunology at Sahlgrenska University Hospital. ACPA was measured using an automated multiplex method (anti-CCP2, BioRad, Hercules, CA, USA). Positive cut-off points were set according to the manufacturer at above 3.0 U/mL. Antibodies against the Fc-region of gamma globulin (total RF) were me asured by rate nephelometric technology (Beckman Immage 800, Beckman Coulter AB, Brea, CA, USA), with a cut-off for RF positivity at 20 U/mL.

### 2.6. Metabolic Parameters

Measurements of the total plasma cholesterol, LDL, and IGF-1 were performed at the accredited Laboratory of Clinical Chemistry at Sahlgrenska University Hospital. The plasma glucose levels were measured using FreeStyle Lite (Abbott Diabetes Care Ltd., Oxon, UK), and insulin levels by sandwich ELISA (DY8056, R&D Systems, Minneapolis, MN, USA), which were used to calculate the homeostatic model assessment for insulin resistance (HOMA-IR) index.

### 2.7. Cytokine Measurement

The measurements of serum IL-4 and IL-10 were performed using sandwich ELISAs (R&D Systems, Minneapolis, MN, USA), and IL-6 (M9316) and IFNγ(M1933) were measured with a sandwich ELISA (Sanquin, Amsterdam, The Netherlands), as previously described [36,37].

### 2.8. Statistical Analysis

The imputation of missing data (<5% for all parameters except for DAS28 (16%) and ACPA (8%)) was performed using the non-parametric R package missForest (version 1.5). Principal component analysis (PCA) was performed in GraphPad Prism (version 9.0) using the following parameters: gender, age, BMI, current smoking, DAS28, positive for RA antibodies and DM antibodies, serum IGF1, total cholesterol, HOMA-IR, plasma insulin, plasma glucose, serum IFNγ, IL10, IL6, and DM at baseline. Binary logistic regression was performed in IBM SPSS (version 28). The first step of the analysis included gender, age, BMI, current smoking, disease duration, MTX dose, DMARD category, DAS28, positive for RA antibodies and DM antibodies, serum IGF1, total cholesterol, HOMA-IR, plasma insulin, plasma glucose, serum IFNγ, IL10, IL4, and IL6. Parameters with *p*-values above 0.1 were eliminated at each step of the analysis. The Kaplan–Meier curves, the Mantel–Cox analysis, the non-parametric Mann–Whitney U-test, and Spearman’s correlation coefficient were all calculated in GraphPad Prism. *p*-values <0.05 were considered statistically significant.

## 3. Results

### 3.1. Characteristics of DM-Related Antibodies

DM-specific antibodies (DMab) were measured in the serums of 290 RA patients. The clinical characteristics of the patients are shown in Figure 2A. We found that 8.3% (27/290) of the patients were positive for at least one DMab. The distribution between different types of DMab is illustrated in Figure 2B. The GADA was the most common DMab (16/27, 59%), followed by ZnT8-ab (8/27, 30%) and IA2-ab (6/27, 22%). A combination of different types of DMab was relatively rare (2/27, 7.4%). The presence of DMab was observed in three out of four patients with T1D (75%) and in two out of fifteen patients with T2D (13%).

The distributions of DMab among the RA patients with and without DM is shown in Figure 2C. DMab was significantly more prevalent in patients with known DM (five out of nineteen vs. twenty-two out of two hundred and seventy-one, OR 4.01 [1.20–11.97], *p* = 0.027). In total, 18.5% of the patients with DMab (five out of twenty-seven) had the disease, compared with only 5.5% in the group with no DMab (fourteen out of two hundred and forty-nine). We observed no statistically significant difference in the frequency of DMab between the male and female RA patients (Figure 2D). However, the prevalence of DM in total and DM with DMab was significantly higher in men (Figure 2D).

### 3.2. Development of New Cases of DM

Next, we investigated the baseline parameters between the patients with DMab and DM. The patients with DMab were younger compared with the DM group (median 48 vs. 56, *p* = 0.045) (Figure 2E). Other clinical parameters, such as disease duration, RA disease activity measured by DAS28, serum cytokine profile, and metabolic profile, were similar between the RA patients with DMab and with prevalent DM.

In the next step, we performed a principal component analysis to identify which of the clinical and serological parameters were associated with DMab (Figure 2F). As is consistent with the results presented above, the PCA confirmed that DMab were significantly associated with DM and metabolic parameters such as BMI, plasma glucose, serum levels of IL6, male gender, and age. Unexpectedly, the production of DMab was also associated with the serum levels of IFNγ and inversely with the serum levels of IGF1. However, we found no significant association between the production of DMab and the RA-specific antibodies (RAab) RF and ACPA, which could potentially be explained by the high prevalence of RAab in the study cohort (Figure 2A,F). Additionally, neither the RA disease activity nor treatment with oral glucocorticoids were associated with production of DMab.

### 3.3. Association of DMab with Metabolic Parameters and Inflammation

Next, we performed the follow-up of the RA patients for development of new cases of DM. Of the total cohort of two hundred and ninety patients, nineteen patients with DM at the baseline were excluded, and an additional six patients (four men and two women) were not available for the follow-up. The remaining 265 patients completed the follow-up that consisted of 2400 patient/years. During this period, 11 new cases of DMs were identified, which increased the prevalence of DM in our study cohort by 61%, from 65.5/1000 patients to 105.6/1000 patients. Only one patient of those eleven new DM cases had measurable DMab at the baseline (Figure 2B). One of twenty-two (4.5%) DMab-positive patients and ten of two hundred and forty-fivepatients (4.08%) without DMab developed DM (HR 1.096, *p* = 0.89) (Figure 3A). Thus, the probability to develop DM was not significantly different between the patients with and without DMab. Unexpectedly, a lack of RAab was associated with a higher probability to develop new cases of DM in our cohort of RA patients (Figure 3A).

New cases of DM developed significantly more often in male than in female RA patients (6/60, and 5/205, *p* = 0.0225). The Kaplan–Meier curve illustrates the difference in the DM-free survival for men and women (Figure 3B), and illustrates a significantly increased estimated risk for new coming DM cases in men (HR 6.08, 95%CI (1.48; 24.9), *p* = 0.014). This outcome demonstrates that in addition to a higher prevalence of DM at the baseline, men were at a high risk for new coming DM in our study cohort.

In the next step, we examined which of the metabolic and inflammatory variables at the baseline were associated with the development of new cases of DM. Patients with DM at the baseline were excluded from this analysis. First, we performed an unsupervised PCA of all the variables and observed that both the inflammatory parameters as disease activity measured by DAS28, as well as the serum levels of RAab, DMab, and IFNγ, were associated with development of new DM (Figure 3C). Notably, the contribution of these inflammatory variables as shown in the PCA analysis was comparable to the traditional metabolic variables which are associated with new DM, including BMI, plasma insulin levels, blood glucose, and HOMA-IR index. To further develop these findings, we performed a multivariate regression analysis of these parameters with the stepwise elimination of the variables that lacked significant association with new DM. At the final step of the analysis, the plasma levels of glucose and insulin and the serum IL6 levels were positively related and the presence of RAab was significantly negatively related to development of new DM in RA patients (Figure 3D).

Due to the fact that both RA and DM are strongly associated with premature mortality [10,11], we investigated how a combination of these diseases affected mortality in our RA cohort. The mortality rate in this study was 16/284 patients per 2400 patient/years of follow-up. Among those death cases, five occurred in patients with DM (five out of thirty, 16.7%), while the remaining eleven ceased patients had no DM (eleven out of two hundred and fifty-four, 4.3%). This corresponded to the 4.4-folds-higher estimated mortality risk for the patients with DM compared with non-diabetic RA patients (OR 4.38 (1.28–13.52), *p* = 0.021) (Figure 2E). Most of the cases of death were connected to DMab-negative DM; however, two out of sixteen (12.5%) deaths occurred in DMab-positive patients. The frequency of death was not different between the DMab positive and DMab-negative RA patients (Figure 2E).

## 4. Discussion

In this study we evaluated an association between autoimmunity and the development of DM among RA patients, and we showed that 20% of RA patients with DM produced DMab. Among the non-diabetic patients, the DMab prevalence was only 9%, indicating that DMab are associated with DM. As an approximate reference, DMab were produced in 10% of patients with T2D and in 0–5% of the normal population [35,38,39]. This suggests that the prevalence of DMab is increased in patients with RA, thus reflecting the autoimmune profiles of these patients.

The patients with DMab did not differ from other RA patients in terms of age, RA disease duration, BMI, or anti-rheumatic treatment. The PCA showed certain collinearity of DMab with plasma glucose and the serum levels of IL6, IGF1, and IFNγ. Moreover, the production of DMab was not associated with gender, RAab RF and ACPA, BMI, or RA disease activity. This bridges DMab production to the immunological background of RA patients. Additionally, the observed connection between DMab and IFNγ is concordant with the pathogenic role of interferons in the pancreatic islet damage and development of T1D in children [40,41] encourages this notion. Two independent Swedish studies demonstrated that patients with DMab-associated T1D had a higher risk of anti-CCP positive RA [42] and of anti-thyroid peroxidase and anti-thyroglobulin positive thyroiditis [43]. In both studies, the diversity of the antigen targets was linked to the genetic backgrounds of the patients carrying the HLA-DRB1 gene and the gene polymorphism in PTPN22. Other studies could not confirm a connection between DM and anti-CCP antibodies [44,45].

We have recently shown that insufficient signaling through the IGF1/insulin receptor could initiate autoimmunity and contribute to RAab production [46]. The inverse relationship observed here between RAab and the new cases of DM is intriguing, and deserves further attention in the coming research. However, in this study cohort, we could not detect any connection between the production of DMab and RAab. This could be partially explained by the observational nature of our study, in which RF/ACPA-positive patients were over-represented, while the number of DMab-positive patients was small.

The study is underpowered to evaluate the risk of DMab for the development of overt DM. Instead, it showed that the risk to develop DM with an autoimmune background of DMab and with a metabolic background could be similar in RA patients. DMab were associated with DM at the baseline, and the ability to detect an association between DMab and new cases of DM was strongly limited by the fact that the male gender turned out to be an important risk factor for DM. Only in five male patients were DMab were detected at baseline; three of those had DM, implicating that only two men with DMab were included in the follow-up. Instead, females accounted for 89% of non-DM cases with DMab. Despite that, new DM cases were less frequent in women, the only new DM case positive for DMab was a woman diagnosed with DM 35 months after the baseline, when the DMab were measured. This argues for the pathogenic potential of DMab.

At baseline, the prevalence of DM in this RA cohort was 6.55%, which corresponded well to 6.8% of DM in the matching age group from a Swedish population study in 2013 [47], and demonstrated no enrichment for DM in our cohort. Similar to other studies, DM in RA was associated with the male gender, which is also coherent with recent reports from the USA, Canada, and Italy [12,14,48]. During the median follow-up period of 9 years, the prevalence of DM increased by 61% and reached 10.6%, which is comparable to the expected increase of prevalent DM in the aged Swedish population in the year 2020 [47]. Importantly, the mortality risk in RA patients with DM was more than four-times higher compared with RA patients without DM.

## 5. Conclusions

Taken together, this study shows that DMab are associated with DM in RA patients. The study is underpowered to draw any conclusions regarding the predictive ability of DMab for new coming DM in RA patients. We observed that a predominant number of the DMab-positive patients remained free from DM after the follow-up period of 10 years. The study attracts attention to the fact that DM comorbidity is associated with a significant increase in the overall mortality of RA patients and requires urgent preventive measures in these patients.

## Figures and Tables

**Figure 1 cells-11-03676-f001:**
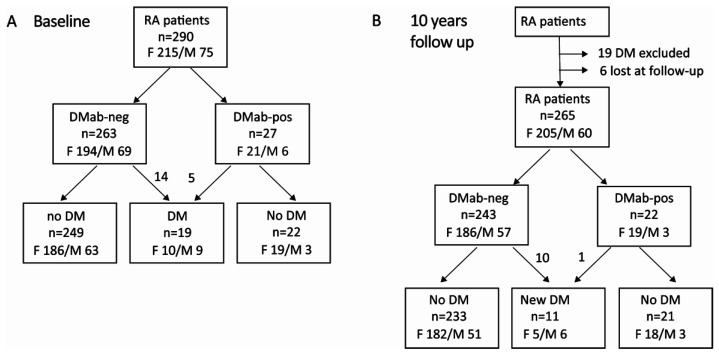
Flow chart of RA patients included in the study. (**A**) Baseline cohort. (**B**) Follow-up cohort. DM, diabetes mellitus; DMab, diabetes-mellitus-specific antibodies; RA, rheumatoid arthritis; M, male; F, female.

**Figure 2 cells-11-03676-f002:**
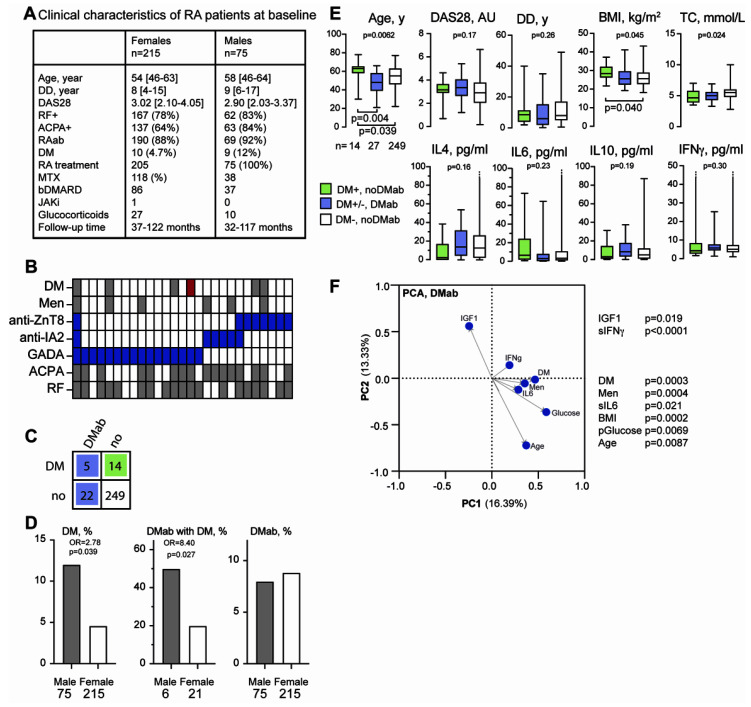
Clinical associates of DM-specific antibodies in RA patients. (**A**) Clinical characteristics of RA patients at baseline. (**B**) Heatmap of DM, rheumatoid factor (RF), antibodies-to-cyclic-citrullinated peptides (ACPA), and gender distribution among the patients with the DM antibodies GADA, anti-ZnT8, and anti-IA2. Red box indicates a case with new DM. (**C**) A 2 × 2 table of coexistence between DM and DMab. (**D**) Bar graphs showing the frequencies of DM, DMab with DM, and DMab in men and women. (**E**) Box plots of clinical and serological variables in patients with DM without DMab (*n* = 14, green), in patients with DMab with and without DM (*n* = 27, blue), and in patients without DM and DMab (*n* = 249, white). (**F**) Scatter plot of principal component analysis (PCA) shows variables associated with DMab. Significantly associated variables are indicated with *p*-value (calculated with least-squares regression). Box plots present median interquartile range, the whiskers indicate min and max values except for when the max is very high (indicated with a line becoming a dotted line). *p*-values were calculated with mixed exact chi-square test and the Kruskal–Wallis test, followed by Dunn’s multiple comparisons test. AU, arbitrary units; BMI, body mass index; DD, disease duration; DM, diabetes mellitus; TC, total cholesterol.

**Figure 3 cells-11-03676-f003:**
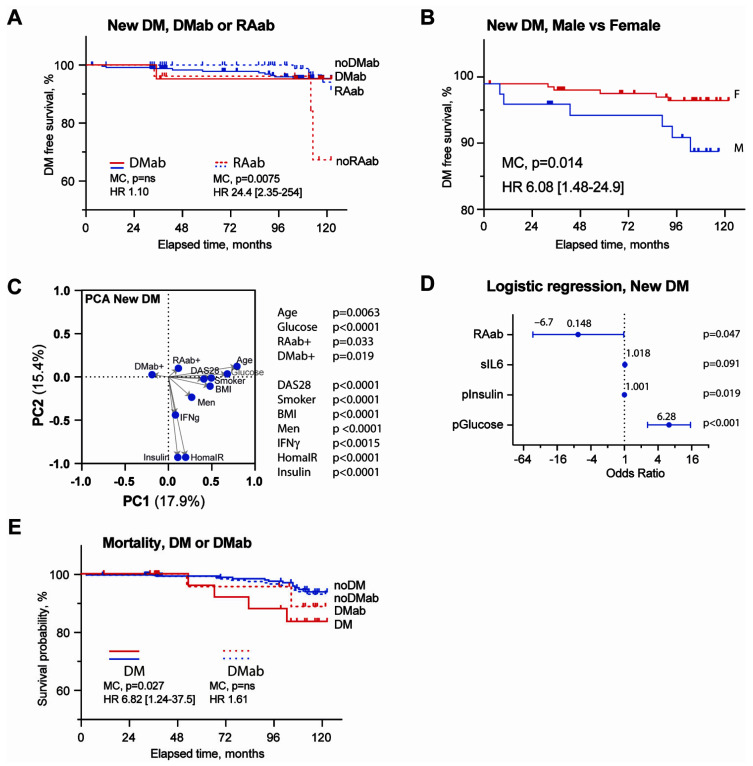
Predictive value of DM-specific antibodies for new cases of DM and mortality in RA patients. (**A**) The Kaplan–Meier curves show the development of new cases of DM in DMab+ (*n* = 27 red line) and DMab-negative (*n* = 257, blue line) patients and in RAab+ (*n* = 234, dash red line) and RAab-negative (*n* = 31, dash blue line) patients during the follow-up period of 120 months. The hazard ratio (HR) between the groups was calculated by the Mantel–Cox (MC) regression analysis. (**B**) The Kaplan–Meier curves show the development of new cases of DM in male (*n* = 66, red line) and female (*n* = 205, blue line) patients. The hazard ratio (HR) between the groups was calculated by the Mantel–Cox regression analysis. (**C**) Scatter plot of principal component analysis (PCA) shows variables associated with new DM cases. Significantly associated variables are indicated with *p*-value (calculated with least-squares regression). (**D**) Forest plot of variables in the final step of the binary logistic regression analysis predicting new DM cases. *p*-values were calculated with Wald statistics. (**E**) The Kaplan–Meier curves show mortality in RA patients with DM (*n* = 30, red line) and without DM (*n* = 254, blue line) and in DMab+ (*n* = 27, dash red line) and DMab-negative (*n* = 257, dash blue line) patients during the follow-up period of 120 months. The hazard ratio (HR) between the groups was calculated by the Mantel–Cox regression analysis.

## Data Availability

All supporting data can be made available upon reasonable request.

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
