# Peer review of "Clinical Significance of Diabetes-Mellitus-Associated Antibodies in Rheumatoid Arthritis"

_cells, 2022, doi:10.3390/cells11223676_

Round 1
Reviewer 1 Report
The manuscript is an interesting piece of work. However, the introduction and discussion can be improved. A few more latest references may be added as some of the references are very old.
Author Response
We changed the wording of the title; we agree it is much clearer. The text of the manuscript is improved. Following suggestions of the referees, 3 references that bind our manuscript to recent studies in the field are added.
We have also modified the conclusion pointing on the urgent need to identify the patients who combine RA with DM. This could be one of the steps to reduce mortality in RA.
Following recommendations of the Technical Editor, we prepared and submit here a graphical abstract of our study to attract immediate attention of the journal readers to the manuscript.
Reviewer 2 Report
Overall, this was an EXTREMELY well written paper! The project design, the experiments, the results, and the conclusion were well documented and explained thoroughly. Also, the subject matter itself was very fascinating and contributes a lot to the field.
The only thing I suggest is to come up with future experiments and drive home why this study is important in the discussion section. Why do we need to know the DMabs in RA patients? What's next after gathering this information?
Other than that, a fantastic job!
Author Response

(The authors gave the same response as above.)
